

# Discovering the differential and gendered consequences of natural disasters on the gender gap in life expectancy in Southeast Asia

Marshal Q. Murillo[1], Shukui Tan[1]

[1]College of Public Administration, Huazhong University of Science and Technology, Wuhan 430074, PR. China

5   *Correspondence to:* Marshal Q. Murillo (murillomarshal@yahoo.com)

**Abstract.** Generally, the country and the population at risk experience the consequences of natural disasters differently and disproportionately. Likewise, the negative impacts of the natural disaster on the population are not gender-neutral. This article explores the relationship of the negative consequences of natural disasters on the difference of gender gap in life expectancy in Southeast Asia. Using the regional data set over the period 1995 to 2011, we analyzed the influence of the natural disaster magnitude, i.e. number of disaster-related casualties, and the interaction with women's socioeconomic and political rights, and the country's vulnerability and exposure to climate change impacts on the gender gap in life expectancy, i.e. ratio of female to male life expectancy. The study produced three important findings. First, Southeast Asian women's life expectancy is more likely to decrease on average compared to that of men as the magnitude of natural disaster increases. Second, lower women's socioeconomic and political conditions are associated with the gender difference in life expectancy as the magnitude of natural disaster increases. Lastly, country's higher level of exposure and vulnerability to climate change impacts are associated with the negative influence of natural disasters on the women's life expectancy more than that of men. Taken together, our study concluded that lower women's socioeconomic and political conditions, as well as country's higher climate change-related vulnerabilities, are likely to pose a collective threat to women's overall well-being more than that of men.

**Keywords.** Natural disaster, gender, women, life expectancy, Southeast Asia, climate change



## 1 Introduction

Over the history, natural disasters have caused significant economic and human losses. Based on the data gathered from the Centre for Research on the Epidemiology of Disaster' (CRED) Emergency Events Database (EM-DAT), various types of natural disaster have claimed more than 32 million lives worldwide since 1900. Environmental shocks such as droughts,

earthquake, cyclones and extreme flood are among the deadliest and disastrous natural disaster types that increased mortality risks and economic losses affecting millions of populations (Gu, Gerland, Pelletier, & Cohen, 2015; Guha-Sapir, Below, & Hoyois, 2016). As exacerbated by climate change phenomenon, natural disasters are expected to affect millions of people across the world and impede current efforts in attaining long-term sustainable development goals (Intergovernmental Panel on Climate Change, 2012; Bergholt & Lujala, 2012; Hallegate, et al., 2016).

There are numerous studies that have explored the adverse consequences of natural disasters on countries and concerned population, particularly the poorest and marginalized groups (Wisner, Blaike, Cannon, & Davis, 2003; Kahn, 2005;  Lin, 2010; Bergholt & Lujala, 2012; Bizzarri, 2012; Bradshaw, 2014; Kousky, 2016). However, there are currently few gender-oriented research works that highlight the direct and disproportionate consequences of catastrophic events on women's life expectancy in relation to the socioeconomic and political conditions present in a country (Neumayer & Plumper, 2007).

Likewise, there is little academic attention given in re-examining the relationship of countries' exposure and capacity to adapt to the impacts of climate change on women's life expectancy.

This article highlights the negative impacts of natural disasters on the gender difference in population's life expectancy in Southeast Asia while examining the interaction effects between two relevant explanatory variables, i.e. women's socioeconomic and political rights, and country's exposure and capacity to adapt to climate change impacts. As such, this

paper contributes to the gender-focused literature that aims to provide a regional understanding of the differential and gendered consequences of natural disasters on women and high-risk countries in Southeast Asia.

The paper proceeds as follows. We started by discussing the theoretical foundation of the impacts of natural disasters that contribute to the disproportionate and unequal challenges experienced by the countries and women at risk. Drawing from previous natural disaster-related and gender-focused literature, we presented our study's main research hypotheses and the

appropriate estimation methodology. We operationalized our proxies for the gender gap in life expectancy, disaster magnitude, women's socioeconomic and political rights, and countries' level of vulnerability and capacity to adapt to climate change impacts. The data results and findings are discussed before we provide our main study's conclusion.

### 1.1 Disproportionate impacts of natural disasters

Natural disasters continuous to affect millions of lives and pose threat to human and economic developments. In Southeast Asia, natural disasters killed more than 500,000 people, accumulated more than 120 billion USD worth of economic damages and affected more than 400 million lives between 1900 to 2016 (Guha-Sapir, Below, & Hoyois, 2016). In a





different report, Kreft and colleagues (2017) summarized the impacts and the socioeconomic costs of natural disasters from the year 1996 to 2015. Of the top ten list of countries that suffered the most from extreme weather-related losses, four Southeast Asian countries were included, i.e. Myanmar, the Philippines, Vietnam, and Thailand. More than eight hundred thousand climate-related deaths were recorded from these four most affected Southeast Asian countries (Kreft, S., Eckstein,

D., & Melchoir, I., 2017). In 2014, the same research group reported that environmental stressors such as extreme and heavy rainfall, widespread flooding, and frequent occurrence of strong typhoons have triggered high a number of deaths and significant economic losses in the countries like Cambodia, Vietnam, Laos, and the Philippines (Kreft, Eckstein, Junghans, Kerestan, & Hagen, 2016).

According to previous studies, the direct negative impacts of natural disasters are not only limited to the economic losses and

high mortality rates. There are other socioeconomic and political situations that are significantly associated with the negative influence of a natural catastrophic event, e.g. the increase of terrorism incidents and civil conflicts at a domestic level (Plumper & Neumayer, 2006; Nel & Righarts, 2008; Berrebi & Ostwald, 2013), and the decrease of fertility rate that hampers the population growth (Lin, 2010).

To understand better why such natural disaster affects the country and the population differently, it is important to identify

how a particular natural phenomenon produces a significant damage. Natural disaster-related research works claimed that natural disasters are distinctly man-made. The exogenous events, like an earthquake, volcanic eruptions, etc., are purely natural while the acquired disaster-related risks are not. For instance, an earthquake that occurred in non-inhabited place cannot be categorized as a disaster because such exogenous occurrence did not affect a particular population and did not cause damages on human and physical properties. From the separate and earlier accounts of Jean-Jacques Rousseau (1756)

and RK Pande (2000), they articulated that people are solely responsible for the human and property losses incurred from a natural disaster (cited in Bose, 2000; cited in Stromberg, 2007). The human actions, decisions, and social conditions of the exposed population, e.g. housing construction standards, urban residence patterns, emergency management protocols, etc., set the level of impact and damages that determine the outcome of a catastrophic event (Stromberg, 2007; Vanderveken & McClean, 2016).

The differential conditions, developments and practices present in a country, such as geography (level of exposure to environmental stressors and natural hazards), poor economic output (income inequality), and institutional quality (presence of free press, governmental accountability and transparency), highly contribute to the differential impacts of natural disaster (Sen, 1999; Freeman, Keen & Mani, 2003; Kahn, 2005; Cohen & Werker, 2008; Donner & Rodriguez, 2008; Collier & Goderis, 2009; Bergholt & Lujala, 2012; Slettebak, 2012; Carlin, Love, & Zechmeister, 2014; Gu, Gerland, Pelletier, &

Cohen, 2015).

Consequently, in face of an equal number of natural disaster, low and middle-income countries suffer more from the nature-induced shocks that lead to higher population mortalities than high-income countries (Gu, Gerland, Pelletier, & Cohen, 2015;





Vanderveken & McClean, 2016). Previous reports and studies illustrated that most victims of natural disasters live in low-income countries, with limited resources to mitigate the risks and damages caused by catastrophic events (Kahn, 2005; Anbarci, Escaleras, & Register, 2005; Nel & Righarts, 2008). According to the report presented by Oxfam International (2013), eighty-six percent of deaths from extreme floods took place in low-lower middle-income countries, ten percent in

upper middle and four percent in high-income countries. Similarly, in a recent report produced by the Centre for Research on the Epidemiology of Disasters (CRED) and UN Office of Disaster Risk Reduction (2016), lower-income countries recorded the highest rate of disaster mortality, with more than 920,000 disaster deaths (sixty-eight percent of total disaster deaths), between 1996 and 2015.

Concomitantly, previous reports and studies theorize that there are likewise opportunities that provide insurance to the

population against the threats of natural disaster. For example, economic developments present in most developed countries help lessen the severity of the negative impacts of a catastrophic event (Kahn, 2005; Bergholt & Lujala, 2012). High-income countries are better suited in establishing natural disaster-related measures, e.g. strong housing fixtures that can withstand an earthquake and extreme flood, structural financial policies, and other disaster-related preparedness and response capacity, that mitigate the unprecedented consequences of natural disasters and limit the mortality risks (Collier & Goderis, 2009;

Schreurs, 2011).

There are academic claims that government and institutional quality also play an important role in insuring the lives of the population before, during and after the occurrence of a disaster. Previous research works linked efficiency, accountability, and transparency to a more democratic government and as a result, political developments lessen the natural disaster-related risks and damages (Besley & Burgess, 2002; Nel & Righarts, 2008). As such, government institutions that underperform and

lack political advances increase the disaster-related risks and vulnerabilities of the exposed population. For instance, according to Escaleras, Anbarci, and Register's (2007) research findings, public sector corruption is positively attributed to earthquake deaths. Similarly, Kahn (2005) argued that those countries that are more democratic and transparent create an opportunity to lessen the disaster-related mortalities.

These collective studies help explain the intersection of socioeconomic and political developments and the negative impacts

of natural disaster. Likewise, these theoretical claims follow the understanding that the unprecedented consequences of natural disasters vary widely. Therefore, there is a reason to believe that socioeconomic and political conditions of the exposed country set the level of risks and vulnerabilities of the exposed population.

### 1.2 Gendered consequences of natural disasters and women's socioeconomic and political status

It is equally important to examine the population at risk and the disproportionate challenges they experience from a natural catastrophic event. This consideration follows the theoretical understanding that there are pre-existing vulnerabilities that largely influence the impacts of natural disasters on particular groups at risk. Wisner and Cannon (1999) consolidated the





definition of vulnerability as the "likelihood that some socially defined group in society will suffer disproportionate death, injury, loss, or disruption of livelihood in an extreme event, or face greater than normal difficulties in recovering from a disaster" (cited in Handmer & Wisner, 1999). For example, poor people tend to settle in hazardous and flood-prone areas because these type of residential spaces are more affordable and accessible regardless of the risks. This specific reality

perpetuates a cycle of disaster for this particular group (Hillier, Oxfam, Nightingale, & Aid, 2013). This disadvantaged group's exposure to environmental shocks like earthquake or storm surge are higher compared to those groups who can afford to settle in a stronger and safer residential space (Cohen & Werker, 2008).

There are reports and studies that illustrated the differential and disproportionate consequences of natural disasters on women. As one of the most marginalized and vulnerable groups in the society, women suffer more from the disproportionate

impacts of natural disasters compared to men (Cannon, 2002; Hillier, Oxfam, Nightingale, & Aid, 2013; Ferris, 2014). In several related case studies, natural disaster-related female casualties are evidently higher than those of men. During the 1991 cyclone in Bangladesh, about ninety-one percent of casualties are women (World Bank, 2012). Women, along with other marginalized groups, were the most affected group when Hurricane Mitch hit Honduras and Nicaragua in 1998 (Nelson, Meadows, Cannon, Morton, & Martin, 2002). Similarly, a high number of women casualties were recorded in Indonesia and

in Sri Lanka following the deadly Indian Ocean tsunami that struck both countries in 2004. In 2008, Myanmar also recorded an estimated sixty-one percent female fatalities after the cyclone hit the country (World Bank, 2012; Alagan & Seela, 2015). This high female mortality rate is linked to systemic socioeconomic, cultural and political marginalization during the onset of a catastrophic event (Begum, 1993; Dankelman, 2002; Cannon, 2002; Donner & Rodriguez, 2008; Aguilar, 2009; Alim, 2009; Habtezion, 2013; Lambrou & Nelson, 2010; Alagan & Seela, 2015). Women in most developing countries are

expected to fulfil the responsibility of looking after their children, the elderly and their family properties, e.g. house, livestock, etc., despite being restricted by social and cultural norms. For example, women are often prohibited to take part in some life-saving activities that are critical during disaster times, like swimming (Cannon, 2002; Nelson, Meadows, Cannon, Morton, & Martin, 2002).

Furthermore, even during post-disaster situations, women continuous to experience unprecedented challenges that either put

their health and well-being at significant risk–e.g. domestic violence, rape, sexual harassment, etc., and even hamper their opportunity to a gainful employment after the occurrence of a disaster–e.g. discrimination in hiring, promotion, and related employment practices (Jenkins & Philipps, 2008; Bradshaw, 2014; Enarson, 2014; David & Enarson, 2012). In addition, it is more difficult for a female-headed household to acquire financial assistance and loans that are essential during the post-disaster rebuilding and re-establishing processes (cited in Alagan & Seela, 2015). Such gender inequalities, unequal burden,

and marginalization present in the society make women more vulnerable than men before, during and after the occurrence of a natural disaster (UN Women, 2016). This gendered consequences of natural disasters lead to higher mortality rate compared to men (Neumayer & Plumper, 2007).



The interconnectedness of the negative impacts of natural disasters on women's life expectancy and the relationship of socioeconomic and political conditions as well as countries' exposure and resiliency against nature's shocks make a strong baseline for our study.

## 5  2 Research hypotheses

As mentioned in the previous section, the occurrence of a natural disaster and its negative impacts implicitly expose the characteristics of a country and the population's socioeconomic and political conditions. This theoretical backdrop provides a good case to re-examine the abovementioned conditions that influence the life expectancy between men and women. Focusing on the Southeast Asian region as our main sample, this study established three research hypotheses for analysis.

In general, the occurrence of a natural disaster may bring negative impact to the population. However, it is the intensity that kills a portion of a population that significantly creates an impact on the population's life expectancy. As articulated in previous literature and studies, women experience inequalities, discrimination, and marginalization in society more than men during the onset of a natural disaster. Comparatively, women are likely to suffer more during and after the onslaught of a natural calamity than men (Nelson, Meadows, Cannon, Morton, & Martin, 2002). Further, in comparison to men's life

expectancy, female mortality is more likely to be adversely affected by the negative impacts of natural disaster as it increases in magnitude (Neumayer & Plumper, 2007).

> *Hypothesis 1: Natural disasters reduce the life expectancy of women relatively more than that of men and the result is likely to increase as the natural disaster strengthens.*

As an institution, the function of the government in providing responsive actions, in the form of enacted laws, plays a key role in safeguarding the population from the threats of natural disasters  (Kahn, 2005; Ferris, 2014). Earlier research pointed out that natural catastrophic events place women in a disadvantaged position making them more susceptible to the adverse consequences of the natural disaster than men. Similarly, in a country where women's social, political and economic rights are well institutionalized, the effect of natural disaster on women's life expectancy relative to that of men diminishes

(Neumayer & Plumper, 2007).

> *Hypothesis 2: Natural disasters reduce the life expectancy of women relatively more than that of men and this effect is more likely to increase in countries with lower women's socioeconomic and political status.*

Identifying the relative importance of country's level of exposure and readiness to combat climatic stressors is vital in

determining the impacts of natural disasters on the population. Furthermore, disaster-focused literature and studies argued that country's characteristics such as geography, national income, institutional capacities to mitigate and adapt to the negative impacts of climate change also play some significant role in ensuring the well-being and security of the population.



For instance, exposure to climate stressors as well as the income equality shields a nation from accumulating natural disaster-related deaths and economic damages (Kahn, 2005; Bergholt & Lujala, 2012).

> *Hypothesis 3: Natural disasters reduce the life expectancy of women relatively more than that of men and this is more likely to increase in countries that are highly vulnerable to the impacts of climate change*

## 3 Research methodology

Drawing from previous gender-focused and natural disaster-related literature, we established three research hypotheses for analysis. The succeeding section provides the operationalization of the study's main variables, i.e. magnitude of natural disaster, gender gap in life expectancy, women's socioeconomic and political rights, and country's level of vulnerability and

exposure to climate change impacts, and the discussion of two interaction effects between our main explanatory variables, i.e. disaster magnitude against women's socioeconomic and political rights; and disaster magnitude against country's level of vulnerability and exposure to climate change shocks.

Using the time-series, cross-national and unbalanced panel datasets from 1995 to 2011, we employ the random effects estimation model with a first-order autoregressive error assumption that resolves the problem of autocorrelation. This

estimation method renders an uncorrelated and unbiased output while allowing variation across our study's main outcome and explanatory variables.

## 4 Data

### 4.1 Gender gap in life expectancy

Data on men and women's life expectancies are taken from International Data Base (IDB) of the U.S. Census Bureau. From this source, data set for both male and female life expectancy are comprehensive, well recorded and have lesser missing data compared to those provided by the World Bank and other data banks (U.S. Census Bureau, 2016).

In measuring the gender gap in life expectancy, we use the ratio of female to male life expectancy instead of the absolute difference. Changes in absolute difference can be associated with men and women's health factors since before a natural

disaster take place, women generally live longer than men (World Health Organization, 2016). Using the absolute difference in life expectancies of men and women implies the possibility that gender gap is still decreasing within the same period even though there is an equal number of deaths between men and women. Therefore, the ratio of female to male life expectancy is a less ambiguous proxy for the gender gap in life expectancy in this study.

### 4.2 Magnitude of natural disaster

The cross-national disaster data are taken from Centre for Research on the Epidemiology of Disasters' (CRED) Emergency Events Database (EM-DAT) collected and made publicly available by the Université Catholique de Louvain's School of



Public Health. EM-DAT contains important core data on the occurrence and effects of more than 22,000 mass disaster, i.e. technological and natural disasters, in the world from 1900 until the present year. The comprehensive database consolidates raw data from various sources, including UN agencies, various international non-governmental organizations, insurance companies, research institutes and press agencies (Guha-Sapir, Below, & Hoyois, 2016).

To qualify as a natural disaster in EM-DAT data recording, at least an event must meet one of the following conditions set by CRED. These include: (1) ten or more people were reported killed; (2) one hundred or more people are reported affected, injured and/or homeless; (3) a country must declare a state of emergency and/or a call for international assistance and support must be made (Guha-Sapir, Below, & Hoyois, 2016). However, due to the limited data available for the study's main variables, i.e. women's socioeconomic and political rights and country's vulnerability to climate stressors index, the research

sample is restricted to the period 1995 to 2011.

For the purpose of this study, we established three important parameters in measuring the magnitude of the natural disaster. First, all types of natural disasters that took place during the sample period are covered. List of disastrous events is restricted to nature-induced disaster types, hence excluding technological and other human-triggered disasters. The measure of natural disasters includes drought, earthquakes, epidemic, flood, insect infestation, landslide, mass movement (dry), storm, volcanic

activity, and wildfire.  Table 1 provides a summary of statistics on disaster types and the corresponding number of occurrences, deaths, and affected the population in Southeast Asia during the sample period.

Second, a country that has reported zero casualties within the study's sample period is excluded from the analysis. It is impossible to estimate the influence of a catastrophic event on the our outcome variable (i.e. gender gap in life expectancy) if a country does not have a unit of measure for disaster magnitude (i.e. number of people killed). Thus, the number of people

killed is necessary in measuring the degree of magnitude of a certain catastrophic event. In addition, a number of deaths are less arbitrary to measure the negative effect of a disaster on the population instead of using the number of affected people as a proxy for the disaster magnitude. Estimates of the number of people killed as a proxy for the natural disaster's magnitude is essential in identifying its impact on the gender gap in life expectancy. Taken all these considerations, we decided to exclude Brunei Darussalam from our sample since the country has a zero natural disaster-related deaths report for the year 1995 to

2011. Table 2 provides the summary of statistics on natural disaster occurrences and the number of deaths in each Southeast Asian country.

Third, to account for the overall measure of disaster magnitude, we normalize the size of a catastrophic event by dividing the number of people killed by the total population of each country. The population per capita data are gathered from the World Bank's database (World Bank, 2016). The use of population per capita is necessary because the influence of natural disasters

on an affected country's life expectancy does not only depend on the degree of disaster strength but also on the population size. All else equal, a country that has a smaller population size will more likely to experience a greater reduction in life



expectancy. The same understanding is employed in considering the gender gap in life expectancy instead of life expectancy itself.

### 4.3 Women's socioeconomic and political rights

To measure the socioeconomic and political rights of women, we use the cross-national assessment data taken from Cingnarelli and Richards' (CIRI) Human Rights Database. The CIRI Human Rights Database contains standard-based measures of several types of internationally-recognized human rights, including physical integrity rights, civil rights, and liberties, workers' and women's rights. The data available on CIRI Human Rights Database are coded based on criteria that reflects "the meanings of various human rights" that are anchored in international human rights law and "to represent the

myriad ways in which the expectations of human rights law and actual government behaviour intersect" (Cingranelli, Richards, & Clay, 2016). The database is created to assist policymakers, researchers, teachers, and students in formulating theories and empirical research about government human rights practices.

To be consistent with the objective of the study, we made three considerations in using CIRI's database women's socioeconomic and political status. First, the necessary variables, i.e. social, economic and political rights, are only available

from 1981 to 2011.

Second, we use CIRI's coded ordinal variables (ranging from 0 to 3) for women's social, economic and political rights. A score of 0 represents a country that lacks legal mechanism in protecting and upholding women's specific rights and women living in this country experience gendered consequences in a given year. In contrast, a score of 3 denotes a country that recognizes and enforces women's specific rights in accordance with the law that prohibits gender discrimination and related

violations. Ideally, we would add all three values and get their corresponding average to account for the combined measure of all women's rights. However, unfortunately, some country-year data are not balanced due to missing and/or incomplete reports. Therefore, as an alternative, we combine all the available country-year measures and get their mean to produce a unit of measurement for women's socioeconomic and political conditions.

Lastly, CIRI has stopped coding the variable for women's social rights since 2009, after which we have no alternative data

for this particular measure. However, to resolve this particular issue, we established an additional estimation model that measures women's economic and political rights to resolve this issue.

### 4.4 Climate change vulnerability index

To account for a country's level of vulnerability to climate stressors, we use the vulnerability index that measures a country's

exposure, sensitivity and capacity to adapt to the negative effects of climate change from University of Notre Dame's Environmental Change Initiative's (ND-ECI) Notre Dame Global Adaptation Index (ND-GAIN). The ND-GAIN country index employs a data-driven approach to illustrate which countries are well equipped and prepared to deal with global





changes brought about by overcrowding, resource constraints, and climate disruptions. In addition, ND-GAIN integrates six 'life-supporting sectors', i.e. food, water, health, ecosystem service, human habitat, and infrastructure in measuring a country's overall vulnerability (Notre Dame Global Adaptation Initiative, 2016).

In our view, ND-GAIN cross-national data provide a comprehensive index compared to all alternatives, because it

summarizes a country's vulnerability to the negative consequences of climate change and other global challenges in addition with the country's readiness to improve resilience. ND-GAIN is institutionalized to assist both private and public sectors in understanding which investments are needed to be prioritized in order to provide a more efficient action and solution to some of the adverse global challenges.

We established two considerations in using the dataset from ND-GAIN. First, ND-GAIN's vulnerability index produces the

computed values from 0 to 1, where a value closer to zero represents a country that is less vulnerable and with high readiness condition to combat the adverse impacts of climate change. Using these original values, we transformed them into our own coded dummy variables (ranging from 1 to 4) to account for the country's level of vulnerability to climatic conditions. A score of 1 represents a country with a high vulnerability and lower readiness to climate change impacts, while a score of 4 denotes a less vulnerable country that has the essential mechanism to cope with the challenges of climate change.

Lastly, unfortunately, the data set available are only for the period of 1995 onward. For the same reason, the study limits the starting period to the year 1995.

### 5 Data analysis and discussion

Table 3 presents two estimates. Model 1 contains our main study's three explanatory variables, i.e. disaster magnitude,

women's socioeconomic and political rights, and country's level of exposure and vulnerability to climate stressors, on the gender gap in life expectancy and two interaction effects between two explanatory variables, i.e. (1) disaster magnitude and women's socioeconomic and political rights, and; (2) disaster magnitude and country's level of vulnerability to climate stressors. In Model 2, we provide an alternative interaction effect that estimates the disaster magnitude against women's economic and political rights. This alternative model excludes the measure for women's social rights.

The estimation of our baseline model, together with two interaction effects, established that all explanatory variables could significantly predict the gender gap in life expectancy. Comparing the two models, the standard error values do not significantly vary from our main estimation output. This renders us an understanding that regardless of the exclusion of women's social rights measure, all explanatory variables have the significant influence on the gender gap in life expectancy.

Based on our first hypothesis, we expect a significantly negative association between the disaster magnitude and the gender

gap in life expectancy. The coefficient value of -0.042 suggests that there is a negative and significant association between the disaster magnitude and the gender gap in life expectancy among our sample. In particular, this reinforces the theoretical



claim that as the natural disaster increase its magnitude, women's life expectancy reduces comparatively more than that of men. This result supports our first hypothesis.

For our second hypothesis, we postulate that lower women's socioeconomic and political rights correlate with the increase of negative effect of natural disasters on the gender gap in life expectancy. The interaction coefficient between the two

explanatory variables, i.e. the disaster magnitude and the level of women's socioeconomic and political rights, has the expected negative sign and size.

The negative coefficient and size of the interaction effects between the two explanatory variables (i.e. -0.101) suggest that the negative influence of natural disasters on the gender gap in life expectancy is associated with lower women's socioeconomic and political rights. Further, in model 2 where we excluded the women's social rights, the estimation output

produced similar predicted probability statistical values with model 1 (i.e. -0.142). Both estimation outcome report that the negative consequences of natural disasters on the gender gap in life expectancy are conditionally linked to Southeast Asian women's social, economic and political developments (or the lack thereof). In particular, a country where women have a higher socioeconomic and political status, the adverse effects of natural disaster on women's life expectancy relative to men diminishes. This phenomenon is opposite in countries where women are not fully benefiting from a higher women's

socioeconomic and political status. Taken together, this outcome validates our second hypothesis.

Lastly, for our third hypothesis, we assume that a country that is less vulnerable and the capacity to mitigate and adapt to climate stressors is less likely to experience the negative effects of natural disasters. In our second interaction estimation, the coefficient has the positive and significant value (i.e. 0.066). This result suggests that a country that has an adequate capacity to mitigate and adapt to climate shocks provides an implicit insurance against the negative impacts of natural disasters on the

gender gap in life expectancy. On the contrary, a country that is more vulnerable to the impacts of climate change is more likely to experience the negative effects of natural disasters and women's life expectancy is disproportionately affected. To simply put, we discover that the negative impacts of natural catastrophes on the gender gap in life expectancy are closely associated with the country's level of exposure, sensitivity, and capacity to adapt to the threats of climate change. Therefore, this finding reinforces our third hypothesis.


### 6 Conclusion

Impacts of natural disasters alter the population's way of life in so many ways. As intensified by the extreme natural catastrophic events and climate change, negative consequences of natural disaster unravel the country's level of vulnerability and other the layers of inequalities present in a population. Such inequalities include less attention to women's

socioeconomic and political conditions.

This article examined the negative influences of natural disaster, women's socioeconomic and political conditions, and the level of vulnerability to the impacts of climate change on the difference in life expectancy between men and women in



Southeast Asia. We have quantitatively illustrated that Southeast Asian women are likely to experience the adverse consequences of natural disasters relatively more than their male counterparts in a given magnitude that causes high death tolls. In addition, lower women's socioeconomic and political developments have a significant influence on the negative impacts of natural disasters on women's life expectancy. Likewise, higher exposure, sensitivity and lower capacity to adapt

5  to the threats of climate change stressors contribute to the disproportionate and negative consequences of natural disasters on the gender gap in life expectancy.

Complimentary to the previous cross-national studies, we highlighted that women's socioeconomic and political conditions in Southeast Asia, as well as the countries' level of exposure and risks from the impacts of climate change, are instrumental in insuring the well-being of women at risk in face of a natural disaster. Our study reinforces both academic and political call

10  to put emphasis in understanding how both sexes experiences natural disasters differently.



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



**Tables**

### Table 1 Summary statistics on natural disasters in Southeast Asia, 1995-2011

| Disaster types | No. of occurrences | No. of deaths | No. of affected population |
|---|---|---|---|
| Drought | 21 | 680 | 38847602 |
| Earthquake | 64 | 184404 | 6564619 |
| Epidemic | 69 | 5785 | 297029 |
| Flood | 324 | 12416 | 91817750 |
| Insect infestation | 1 | 0 | 200 |
| Landslide | 64 | 3459 | 685673 |
| Mass movement (dry) | 1 | 11 | 0 |
| Storm | 203 | 156735 | 83284538 |
| Volcanic activity | 26 | 329 | 585829 |
| Wildfire | 15 | 245 | 34300 |
| **Total** | **788** | **364064** | **222117540** |

### Table 2 Statistics on natural disasters per country, 1995-2011

| Country | No. of occurrences | No. of deaths |
|---|---|---|
| Brunei Darussalam | 1 | 0 |
| Cambodia | 26 | 1558 |
| Indonesia | 214 | 184779 |
| Laos | 17 | 233 |
| Malaysia | 53 | 841 |
| Myanmar | 22 | 139359 |
| Philippines | 248 | 15611 |
| Singapore | 3 | 36 |
| Thailand | 78 | 11265 |
| Timor-Leste | 8 | 27 |
| Vietnam | 118 | 10355 |





**Table 3 Main estimation results**

**Dependent variable:**
**Gender gap in life expectancy**

|  | Model 1 | Model 2 |
|---|---|---|
| **Disaster death per hundred thousand people** | - 0.042 (0.013)** | - 0.051 (0.018)** |
| **Women's socioeconomic and political rights** | 0.011 (0.013) | 0.011 (0.013) |
| **Country's vulnerability to climate stressors** | 0.055 (0.015)** | 0.067 (0.019)** |
| **Interaction effect: disaster magnitude x women's socioeconomic and political rights** | - 0.101 (0.043)* | |
| **Interaction effect: disaster magnitude x country's vulnerability to climate stressors** | 0.066 (0.019)** | 0.085 (0.023)** |
| **Interaction effect: Disaster magnitude x women's political and economic rights** | | - 0.142 (0.052)** |
| **No. of observations** | 163 | 170 |
| **No. of countries** | 10 | 10 |
| **R-square** | 0.52 | 0.53 |
| **Wald chi-square** | 153.76** | 173.61** |

*Important notes: Gender gap in life expectancy is the dependent variable. Standard errors are in parentheses. Model 1 is the baseline model of the study that includes two interaction effects (i.e. disaster magnitude against women's socioeconomic and political rights, and disaster magnitude against country's vulnerability index). Model 2 presents an alternative interaction effect that estimates women's economic and political rights against disaster magnitude.*
*$**p<0.01$ (two-sided z-test)*
*$*p<0.05$*



**Data availability**

These datasets were generated from the following public domain resources:

1. Data on men and women's life expectancies (1995-2011):

   5     https://www.censuss.gov/population/international/data/idb/informationGateway.php

2. Data on country's population (1995-2011): https://data.worldbank.org/indicator/SP.POP.TOTL?page=2

3. Data on natural disaster-related occurrences and mortalities*: http://emdat.be/emdat_db/

4. Data on women's socioeconomic and political rights: http://www.humanrightsdata.com/p/data-documentation.html

5. Data on country's climate change vulnerability index: http://index.gain.org/ranking/vulnerability

*Important note: The data on natural disaster-related occurrences and mortalities provided by the EM-DAT requires registration prior to academic and/or private use. However, the researcher has provided the dataset, through a table form, on this article (see Table 2).