# Peer review of "Discovering the differential and gendered consequences of natural disasters on the gender gap in life expectancy in Southeast Asia"

_Natural Hazards and Earth System Sciences, 2017_

## Referee Comment (RC1) · O. Petrucci (Referee) · 28 Dec 2017

The paper "Discovering the differential and gendered consequences of natural disasters on the gender gap in life expectancy in Southeast Asia" focuses on the gender differences in case of natural disasters in Southeast Asia. The Authors tested a series of research hypotheses and finally concluded that during natural disasters females are more vulnerable than males in terms of life expectancy. I think that the paper is well organised and clear, but I think that two points should be necessarily clarified.

[Figure]

1) The paper is based on the data collected in EM-DAT, particularly on the number of fatalities caused by natural disasters. I think that there is a basic question to point out. EM-DAT collects all types of natural disasters, no matter their triggering causes. The authors mentioned the climate change in different parts of the paper and talks about the possible modifications of risk for human life related to the increasing impacts of natural disasters due to the climate change. The "Hypothesis 3: Natural disasters reduce the life expectancy of women relatively more than that of men and this is more likely to increase in countries that are highly vulnerable to the impacts of climate change" as formulated, implies that "all" the natural disasters can be affected by climate change. Nevertheless, several natural disasters collected in EM-DAT are not related to climate, thus cannot be affected by climate change. Earthquakes and volcanic disasters have an endogenous trigger, while mass movements (landslides is a type of mass movements) (Page 8 line 14) can be either related to climatic features or linked to earthquakes. Thus, some natural disasters are not connected to climate and cannot be directly related to climate change. Then I think that this basic point must be clarified. The disasters cannot be considered all together.

2) I understand that the paper concerns a specific area (Southeast Asia), but I think that the Authors should also mention studies carried out in developed countries where the vulnerability is larger for males than for females. I think that the Authors should give visibility to these opposite findings obtained in different countries. Otherwise, a generic reader can believe that the findings of this paper are verified in all the countries.

---

## Author Comment (AC1) · 14 Mar 2018

We would like to thank the reviewer for her remarks and comments. We have addressed the following points and they are all well taken.

Attached the revised copy of the article, for reference and further scrutiny.

POINT No. 1: "The paper is based on the data collected in EM-DAT, particularly on the number of fatalities caused by natural disasters. I think that there is a basic question to point out. EM-DAT collects all types of natural disasters, no matter their triggering

causes. The authors mentioned the climate change in different parts of the paper and talks about the possible modifications of risk for human life related to the increasing impacts of natural disasters due to the climate change.

The "Hypothesis 3: Natural disasters reduce the life expectancy of women relatively more than that of men and this is more likely to increase in countries that are highly vulnerable to the impacts of climate change" as formulated, implies that "all" the natural disasters can be affected by climate change.

Nevertheless, several natural disasters collected in EM-DAT are not related to climate, thus cannot be affected by climate change. Earthquakes and volcanic disasters have an endogenous trigger, while mass movements (landslides is a type of mass movements) (Page 8 line 14) can be either related to climatic features or linked to earthquakes. Thus, some natural disasters are not connected to climate and cannot be directly related to climate change. Then I think that this basic point must be clarified. The disasters cannot be considered all together."

REPLY: Point noted. The third hypothesis is rephrased to "natural disasters reduce the life expectancy of women relatively more than that of men, and this is more likely to increase in countries that are highly vulnerable and exposed to nature-induced shocks."

The measures for the "country's level of vulnerability and exposure to climate- and nature-induced shocks" are collected in Notre Dame Global Adaptation Initiative (ND-GAIN). This collected data measures the country's overall exposure, sensitivity, and vulnerability regarding "six life-supporting sectors - food, water, health, ecosystem service, human habitat, and infrastructure." We believe that the ND-GAIN data represents the country's vulnerability and exposure to nature-induced shocks, regardless if a natural disaster is climatological (e.g. drought, wildfire, etc.), hydrological (e.g., flood, landslide, etc.), or geophysical (e.g., earthquake, mass movement, etc.). The cross-national data is also a comprehensive index that measures the country's capacity to address not only the impacts of climate change but also "other global challenges."

POINT NO. 2: "I understand that the paper concerns a specific area (Southeast Asia), but I think that the Authors should also mention studies carried out in developed countries where the vulnerability is larger for males than for females. I think that the Authors should give visibility to these opposite findings obtained in different countries. Otherwise, a generic reader can believe that the findings of this paper are verified in all the countries."

– Point noted. The revised study provides an opposing view on the matter. That is to say; we find no significant evidence to support that the study's two determinants (i.e., women's socio-economic and political rights and the country's level of vulnerability to natural catastrophic events) exacerbate the life expectancy of both women and men in the face of natural disaster.

Also, as suggested, we added few literature and studies that showcased men's vulnerability to the impacts of natural disaster. See page 6, line 10-15.

We thank the reviewer for her time and as well as her careful review of our article.

Please also note the supplement to this comment:
https://www.nat-hazards-earth-syst-sci-discuss.net/nhess-2017-370/nhess-2017-370-AC1-supplement.pdf

**Supplement:**

**Discovering the differential and gendered consequences of natural disasters on the gender gap in life expectancy in Southeast Asia**

Shukui Tan[1], Marshal Q. Murillo[1]

[1]College of Public Administration, Huazhong University of Science and Technology, Wuhan 430074, PR. China

5   *Correspondence to:* Marshal Q. Murillo (murillomarshal@yahoo.com)

**Abstract.** The country and the population at risk experience the negative impact of natural disasters differently and disproportionately. This article explores the influence of the threats of natural disasters on the difference of gender gap in life expectancy in Southeast Asia. Using the regional data set over the period 1995 to 2011, we estimated the influence of the

10   natural disaster strength, i.e., number of disaster-related casualties, and its interaction with women's socio-economic and political rights, and the country's vulnerability and exposure to climate- and nature-induced shocks on the gender gap in life expectancy, i.e., ratio of female to male life expectancy. The findings showed that on average, Southeast Asian women's life expectancy is more likely to decrease compared to that of men as the magnitude of natural disaster increases. However, women's socio-economic and political rights, as well as the country's vulnerability to natural catastrophic stressors are not

15   predictive of the gender gap in life expectancy in the face of natural disaster. The findings support the widespread perception that the impact of natural disaster is not gender neutral, however, some socio-economic and political advancements in Southeast Asia do not significantly contribute to the gendered and disproportionate impact of natural catastrophic events on women's life expectancy.

20   **Keywords:**      Natural disaster, gender, vulnerability, life expectancy, Southeast Asia

**1. Introduction**

Over the history, natural disasters have caused significant economic and human losses. Based on the data gathered from the Centre for Research on the Epidemiology of Disaster' (CRED) Emergency Events Database (EM-DAT), various types of natural disaster have claimed more than 32 million lives worldwide since 1900. Environmental shocks such as droughts, earthquake, cyclones and extreme floods are among the deadliest and disastrous natural disaster types that increased mortality risks and economic losses affecting millions of populations (Gu, Gerland, Pelletier, & Cohen, 2015; Guha-Sapir, Below, & Hoyois, 2016). As exacerbated by climate change phenomenon, natural disasters are expected to affect millions of people across the world and impede current efforts in attaining long-term sustainable development goals (IPCC, 2012; Bergholt & Lujala, 2012; Hallegate et al., 2016).

There are growing number of cross-country studies that explored the adverse consequences of natural disasters on population at risk, particularly underscoring the impact on most deprived and marginalized groups (Wisner, Blaike, Cannon, & Davis, 2003; Kahn, 2005; Lin, 2010; Bergholt & Lujala, 2012; Bizzarri, 2012; Bradshaw, 2014; Kousky, 2016). However, there are currently few gender-oriented research works that highlight the direct and disproportionate consequences of catastrophic events on women's life expectancy and linking this phenomenon to the socio-economic and political conditions present in most countries (Neumayer & Plümper, 2007; Clark & Peck, 2012). Likewise, there is little academic attention given in re-examining the relationship of countries' exposure and capacity to adapt to the impacts of nature-induced shocks on women's life expectancy in Southeast Asia – one of the most vulnerable regions against natural catastrophic events.

This research highlights the negative impact of natural disasters on the gender difference in population's life expectancy in Southeast Asia while examining the interaction effects between two relevant explanatory variables, i.e., women's socio-economic and political rights, and the country's exposure and capacity to adapt to the negative impacts of natural catastrophic events. As such, this article contributes to the gender-focused literature that aims to provide a regional perspective of the differential and gendered consequences of natural disasters on women and high-risk countries in Southeast Asia.

The paper proceeds as follows. we started by discussing the theoretical foundation of the impacts of natural disasters that contribute to the disproportionate and unequal challenges experienced by the countries and population at risk. Drawing from previous natural disaster-related and gender-focused literature and reports, we presented the study's principal research hypotheses and discussed the appropriate estimation methodology. Also included are the operationalized unit of measures for the gender gap in life expectancy, disaster magnitude, women's socio-economic and political rights, and countries' level of vulnerability and capacity to adapt to the threats of climate- and nature-induced stressors. Finally, the article ends with a summary and conclusion.

**1.1 Disproportionate impacts of natural disasters**

Natural disasters continue to affect millions of lives and pose a threat to human and economic developments. In Southeast Asia, natural disasters killed more than 500,000 people, accumulated more than 120 billion USD worth of economic damages and affected more than 400 million lives between 1900 to 2016 (Guha-Sapir, Below, & Hoyois, 2016). In a different report, Kreft and colleagues (2014) summarized the impacts and the socio-economic costs of natural disasters from the year 1994 to 2013. Of the top ten list of countries that suffered the most from extreme weather-related losses, three Southeast Asian countries were included, i.e., Myanmar, Vietnam, and the Philippines. The same report shows environmental stressors such as intense and heavy rainfall, widespread flooding, and frequent occurrence of strong typhoons have triggered high a number of deaths and significant economic losses in the countries like Cambodia, Vietnam, Laos, and the Philippines (Kreft, Eckstein, Junghans, Kerestan, & Hagen, 2014).

Previous studies demonstrated that the direct adverse impacts of natural disasters are not only limited to the economic losses and high mortality rates. There are other socio-economic and political situations that are significantly associated with the adverse consequence of a natural catastrophic event, e.g. the increase of terrorism incidents and civil conflicts at a domestic level (Plümper & Neumayer, 2006; Nel & Righarts, 2008; Berrebi & Ostwald, 2013), decrease of social trust (Albrecht, 2017), corruption in public sector (Yamamura, 2014), and even the decrease of fertility rate that affects the population growth (Lin, 2010).

To better understand why such natural disaster affects the country and the population differently, it is essential to identify how a natural catastrophic event inflicts significant damage. Related research works claimed that natural disasters are distinctly human-made. The exogenous events, like an earthquake, volcanic eruptions, etc., are inherently natural while the acquired disaster-related risks are not. For instance, an earthquake that occurred in non-inhabited place cannot be categorized as a disaster because such exogenous occurrence did not affect a particular population and did not cause damage to human and other physical properties. From the separate and earlier accounts of Jean-Jacques Rousseau (1756) and RK Pande (2000), they articulated that people are solely responsible for the human and property losses incurred from a natural disaster (cited in Bose, 2000; cited in Stromberg, 2007). The human actions, decisions, and social conditions of the exposed population, e.g., housing construction standards, urban residence patterns, emergency management protocols, etc., set the level of impact and damages that determine the adverse outcome of a nature-induced catastrophic event (Stromberg, 2007; Vanderveken & McClean, 2016).

The differential conditions, developments and practices present in a country, such as geography (level of exposure to environmental stressors and natural hazards), weak economic output (income inequality), and institutional quality (presence of free press, governmental accountability and transparency), significantly contribute to the unequal challenges and burden caused by a natural disaster (Freeman, Keen, & Mani, 2003; Kahn, 2005; Cohen & Werker, 2008; Donner & Rodriguez, 2008; Collier & Goderis, 2009; Bergholt & Lujala, 2012; Slettebak, 2012; Carlin, Love, & Zechmeister, 2014; Gu, Gerland, Pelletier, & Cohen, 2015). Consequently, in the face of an equal number of natural disaster, low and middle-income natios

suffer more from the nature-induced shocks that lead to higher population mortalities than high-income countries (Gu, Gerland, Pelletier, & Cohen, 2015; Vanderveken & McClean, 2016). Previous reports and studies revealed that most victims of natural disasters live in low-income countries, with limited resources to mitigate the risks and damages inflicted by natural catastrophic events (Kahn, 2005; Escaleras, Anbarci, & Register, 2005; Nel & Righarts, 2008; Hallegate, Vogt-Schilb, Bangalore, & Rozenberg, 2017). In particular, according to the report presented by Oxfam International (2013), eighty-six percent of deaths from extreme floods took place in low-lower middle-income countries, ten percent in upper middle and four percent in high-income countries. Similarly, according to the data released the Centre for Research on the Epidemiology of Disasters (CRED) and UN Office of Disaster Risk Reduction (2016), lower-income countries recorded the highest rate of disaster mortality, with more than 920,000 disaster deaths (sixty-eight percent of total disaster deaths), between 1996 and 2015.

Concomitantly, previous reports and studies theorize that there are likewise opportunities that provide insurance to the population against the brunt of natural disaster. For example, economic developments present in most developed countries help lessen the severity of the negative impacts of a catastrophic event (Kahn, 2005; Bergholt & Lujala, 2012; Hallegate, Vogt-Schilb, Bangalore, & Rozenberg, 2017). High-income countries are better suited in establishing natural disaster-related measures through sturdy housing fixtures that can withstand an earthquake and extreme flood, structural financial policies, and other disaster-related preparedness and response capacity, that mitigate the unprecedented consequences of natural disasters and limit the mortality risks (Collier & Goderis, 2009; Schreurs, 2011).

Government and institutional quality play an important role in ensuring the lives of the population before, during and after the occurrence of a disaster. Previous research works linked efficiency, accountability, and transparency to a more democratic government and as a result, political developments lessen the natural disaster-related risks and damages (Besley & Burgess, 2002; Burby, 2006; Nel & Righarts, 2008). As such, government institutions that underperform and lack political advances increase the disaster-related risks and vulnerabilities, as well as the changes in public opinion and corrosion of social trust of the exposed population (Carlin, Love, & Zechmeister, 2014; Albrecht, 2017). For instance, according to Escaleras, Anbarci, and Register's (2007) research findings, public sector corruption is positively attributed to earthquake deaths. Similarly, Kahn (2005) argued that those countries that are more democratic and transparent create an opportunity to lessen the disaster-related mortalities.

These studies provide a theoretical background in understanding the interaction between the socio-economic and political features present in a country and the deadly threats of natural disaster. Likewise, socio-economic and political conditions of the exposed country set the level of risks and vulnerabilities of the exposed population.

**1.2 Gendered consequence of natural disaster**

It is equally important to examine the population at risk and the disproportionate challenges they experience from a natural catastrophic event. This consideration follows the theoretical understanding that there are pre-existing vulnerabilities that largely shape the impact of natural disasters on particular groups. Wisner and Cannon (1999) consolidated the definition of vulnerability as the "likelihood that some socially defined group in society will suffer disproportionate death, injury, loss, or disruption of livelihood in an extreme event, or face greater than normal difficulties in recovering from a disaster" (cited in Handmer & Wisner, 2010). For example, poor people tend to settle in hazardous and flood-prone areas because these type of residential spaces are more affordable and accessible regardless of the risks. This specific reality perpetuates a cycle of disaster for this particular group (Hillier, Oxfam, Nightingale, & Aid, 2013). For instance, the exposure of a particularly disadvantaged group to environmental shocks like earthquake or storm surge is higher compared to those groups who can afford to settle in a stronger and safer residential space (Cohen & Werker, 2008).

In the context of gender vulnerability, women and men belonging to different socio-economic strata experience distinct risk and vulnerabilities to the impacts of natural disaster (Habtezion, 2013). There are numerous reports and studies that illustrated the distinct consequences of a natural disaster experienced by women relative to their male counterpart. As one of the most marginalized and vulnerable groups in the society, women suffer more from the disproportionate impacts of natural disasters compared to men (Cannon, 2002; Hillier, Oxfam, Nightingale, & Aid, 2013; Ferris, 2014). For instance, in a study of Irshad et al., (2012), they reported the long-term gendered of the 2005 Pakistan earthquake on disabled women. The study discovered that paraplegic women are evidently marginalized in terms of social, emotional, and financial context (Irshad, Mumtaz, & Levay, 2011).

Likewise, in several related case studies, natural disaster-related female fatalities are evidently higher than those of men. During the 1991 cyclone in Bangladesh, about ninety-one percent of casualties are women (World Bank, 2012). Women, along with other marginalized groups, were the most affected group when Hurricane Mitch hit Honduras and Nicaragua in 1998 (Nelson, Meadows, Cannon, Morton, & Martin, 2002). Similarly, a high number of women casualties were recorded in Indonesia and Sri Lanka following the deadly Indian Ocean tsunami that struck both countries in 2004. In 2008, Myanmar also recorded an estimated sixty-one percent female fatalities after the cyclone hit the country (World Bank, 2012; Alagan & Seela, 2015).

This high female mortality rate is linked to systemic socio-economic, cultural and political marginalization during the onset of a catastrophic event (Begum, 1993; Dankelman, 2002; Cannon, 2002; Donner & Rodriguez, 2008; Aguilar, 2009; Alim, 2009; Habtezion, 2013; Lambrou & Nelson, 2010; Alagan & Seela, 2015). Women in most developing countries are expected to fulfill the responsibility of looking after their children, the elderly and their family properties, e.g., house, livestock, etc., despite being restricted by social and cultural norms. For example, women are often prohibited to take part in some life-saving activities that are critical during disaster times, like swimming, climbing trees, etc. (Cannon, 2002; Nelson, Meadows, Cannon, Morton, & Martin, 2002; Peek & Fothergill, 2008).

Furthermore, even during post-disaster situations, women continuous to experience unprecedented challenges that either put their health and well-being at significant risk–e.g. domestic violence, rape, sexual harassment, etc., and even hamper their opportunity to a gainful employment after the occurrence of a disaster–e.g. discrimination in hiring, promotion, and related employment practices (Jenkins & Philipps, 2008; Bradshaw, 2014; Enarson, 2014; David & Enarson, 2012). In addition, it is more difficult for a female-headed household to acquire financial assistance and loans that are essential during the post-disaster rebuilding and re-establishing processes (cited in Alagan & Seela, 2015). Such gender inequalities, unequal burden, and marginalization present in the society make women more vulnerable than men before, during and after the occurrence of a natural disaster (UN Women, 2016). Thus, the overall socio-economic and political conditions of women significantly lead to higher mortality rate compared to men (Neumayer & Plümper, 2007).

Likewise, men bore the negative consequences of natural disasters too. In particular, there are studies that illustrate how men are more likely to be vulnerable to flood- and storm-related disasters than their female counterpart in most medium and high-income countries (Ashley & Ashley, 2008; Yeo & Blong, 2010). In a study by Zagheni et al. (2015), they found that male deaths were higher than that of the female during the storm- and flood events. This phenomenon is likely to be rooted in the risk-taking behavior that is commonly attributable to male adults than their female counterpart (Kruger & Nesse, 2005; Bradshaw, 2004).

The theorized influences of the negative impacts of natural disasters on life expectancy, as well as the interaction of a country's socio-economic and political conditions and level of vulnerability to natural catastrophic events, make this study imperative and relevant.

**2. Research hypotheses**

As mentioned in the previous chapter, the occurrence of a natural disaster and its negative impacts implicitly expose the vulnerability of a country and the population's socio-economic and political conditions. This theoretical backdrop provides a relevant case to re-examine the socio-economic and political conditions that influence the life expectancy of women and men. Focusing on the Southeast Asian region as the main sample, this study established three research hypotheses for analysis.

Fundamentally, the occurrence of a natural disaster may bring negative impact to the population. However, it is the intensity that kills a portion of a community that significantly creates an effect on the population's life expectancy. As articulated in previous literature and studies, women experience inequalities, discrimination, and marginalization in society more than men during the onset of a natural disaster. Comparatively, women are likely to suffer more during and after the onslaught of a natural calamity than men (Nelson, Meadows, Cannon, Morton, & Martin, 2002; Hillier & Nightingale, 2013). Further, relative to their male counterpart, female mortality is more likely to be adversely affected by the negative impacts of natural disaster as its strength increases (Neumayer & Plümper, 2007).

**Hypothesis 1.** Natural disasters reduce the gender gap in life expectancy, i.e., ratio of female to male life expectancy

As an institution, the function of the government in providing responsive actions, in the form of enacted laws and established macro-level policies, plays a crucial role in safeguarding the population from the threats of natural disasters (Kahn, 2005; Burby, 2006; Ferris, 2014). Earlier research works pointed out that natural catastrophic events place women in a disadvantaged position making them more susceptible to the adverse consequences of the natural disaster than men. Similarly, in a country where women's social, political and economic rights are well institutionalized, the effect of natural disaster on women's life expectancy relative to that of men diminishes (Neumayer & Plümper, 2007).

**Hypothesis 2.** Natural disasters reduce the life expectancy of women relatively more than that of men, and this effect is more likely to increase in countries with lower women's socio-economic and political status.

Identifying the role of country's level of exposure and vulnerability to climate- and nature-induced stressors is vital in determining the impacts of natural disasters on the population. Furthermore, disaster-focused literature and studies argued that country's characteristics such as geography, national income, institutional capacities to mitigate and adapt to the negative impacts of natural catastrophic events likewise play some significant role in reducing the risk of the exposed population. For instance, exposure to climate stressors as well as the income equality shields a nation from accumulating natural disaster-related deaths and economic damages (Kahn, 2005; Bergholt & Lujala, 2012).

**Hypothesis 3.** Natural disasters reduce the life expectancy of women relatively more than that of men, and this is more likely to increase in countries that are highly vulnerable and exposed to climate- and nature-induced shocks

**3. Research methodology**

Drawing from previous gender-focused and natural disaster-related literature, we established three research hypotheses for testing and analysis. The succeeding section provides the operationalization of the study's primary variables, i.e., magnitude of natural disaster, gender gap in life expectancy, women's socio-economic and political rights, and country's level of vulnerability and exposure to the impacts of natural catastrophic events, and the discussion of two interaction effects between the primary explanatory variables, i.e., disaster magnitude against women's socio-economic and political rights; and disaster magnitude against country's level of vulnerability and exposure to climate- and nature-induced shocks.

Using the time-series, cross-national and unbalanced panel datasets from 1995 to 2011 (n=300), this study tested and analyzed three established hypotheses through different estimation methods, i.e., moderation analysis, hierarchical multiple regression, and three-way moderation estimation

**4. Data**

**4.1. Gender gap in life expectancy**

[revised manuscript text omitted]

that recognizes and enforces women's specific rights by the law that prohibits gender discrimination and related violations. Ideally, we would add all three values and get their corresponding average to account for the combined measure of all women's rights. However, unfortunately, some country-year data are not balanced due to missing and incomplete reports. Therefore, as an alternative, we combine all the available country-year measures and get their mean to produce a unit of measurement for women's socio-economic and political conditions.

Lastly, CIRI has stopped coding the variable for women's social rights since 2009, after which we have no alternative data for this particular measure.

**4.4. Country's vulnerability to natural- and climate-induced stressors**

To account for a country's level of vulnerability to climate- and nature-induced stressors, we use the vulnerability index that measures a country's exposure, sensitivity and capacity to adapt to the adverse effects of natural catastrophic shocks from University of Notre Dame's Environmental Change Initiative's (ND-ECI) Notre Dame Global Adaptation Index (ND-GAIN). The ND-GAIN country index employs a data-driven approach to illustrate which countries are well equipped and prepared to deal with global changes brought about by overcrowding, resource constraints, and climate disruptions. In addition, ND-GAIN integrates six 'life-supporting sectors,' i.e., food, water, health, ecosystem service, human habitat, and infrastructure in measuring a country's overall vulnerability (ND-GAIN, 2016).

In this view, ND-GAIN cross-national data provide a comprehensive index compared to all alternatives, because it summarizes a country's vulnerability to the negative consequences of climate- and nature-induced events and other global challenges also with the country's readiness to improve resilience. Moreover, ND-GAIN is institutionalized to assist both private and public sectors in understanding which investments are needed to be prioritized to provide a more efficient action and solution to some of the adverse global challenges.

We established two considerations in using the dataset from ND-GAIN. First, ND-GAIN's vulnerability index produces the computed values from 0 to 1, where a value closer to zero represents a country that is less vulnerable and with high readiness condition to combat the adverse impacts of climate- and nature-induced events and other global challenges. Using these original values, we transformed them into the study's coded dummy variables (ranging from 1 to 4) to account for the country's level of vulnerability to climate- and nature-induced stressors. A score of 1 represents a state with high vulnerability and lower readiness, while a score of 4 denotes a less vulnerable country that has the essential mechanism to cope with the threats mentioned above.

Lastly, unfortunately, the data set available are only from 1995 onward. For the same reason, the study limits the starting period to the year 1995.

**5. Research findings**

**5.1. Data analysis**

In testing the study's four established hypotheses, three different estimation methods are employed, i.e. hierarchical multiple regression and three-way moderation analysis. These methods will be discussed individually in the succeeding paragraph.

To produce efficient estimates and unbiased results, it is necessary to resolve two classical statistical problems, i.e. the outliers in our data and the issue of multicollinearity. There are three identified outliers, i.e. Indonesia (2004), Myanmar (2008), and Thailand (2004). These identified outliers were deducted from the analysis. In addition, the pre-estimation result shows that the study's independent variables are not strongly correlated, hence no case of multicollinearity.

**Hypothesis 1.** Magnitude of natural disaster reduces the gender gap in life expectancy

The first hypothesis aims to examine whether the magnitude of natural disaster reduces the gender gap in life expectancy (see figure 1). To test the first hypothesis, the linear and quadratic curve are inspected. Figure 2 reports that there is one bend in the regression line. In order to inspect whether there is a quadratic effect, hierarchical regression approach is conducted.

Results in table 3 show the coefficient of determination for both linear and quadratic function. R-square value for linear regression indicates that magnitude of natural disaster explains 0.96 percent of the variability of the life expectancy. R-square value of the quadratic function shows that magnitude of natural disaster explains 0.173 percent of the variability of the life expectancy. In addition, the results show that there is a significant improvement in the model when the squared variable is added ($p=0.000$, $p<0.05$).

F-ratio in Table 4 tests whether the overall regression model is a good fit for the data. In quadratic function, the results show that the independent variable statistically and significantly predicts the dependent variable $F(2,147)=15.334$, $p=0.000$, $R^2=0,173$. Also, Table 4 confirms that quadratic regression equation can be used to predict y-values. Where $Y$ is equal to the gender gap in life expectancy (i.e. female life expectancy over male's life expectancy) and $X$ is equal to the magnitude of natural disaster (death over the population per capita x 100,000).

$$Y = \beta_0 + \beta_1 * X_1 + \beta_2 * X_1^2$$

In table 5, the results show that gender gap at low levels of the magnitude of natural disaster is increasing and decreases at the certain level of the magnitude of the natural disaster. Based on these, we can conclude that magnitude of natural disaster significantly reduces the gender gap in life expectancy after a certain level of the natural disaster ($p=0.000$, $p<0,05$). Put differently, as the magnitude of natural disaster increases, the gender gap in life expectancy also increases until a certain level of natural disaster strength. After reaching such level, the gender gap in life expectancy starts to decrease. Therefore, we will accept the first hypothesis.

**Hypothesis 2.** Magnitude of natural disaster reduces the life expectancy of women relatively more than that of men and this is more likely to be more evident in countries with lower women's socio-economic and political rights

In table 6, model 2 reports that gender significantly predicts the life expectancy, b=5.0617, t(292)=6.3736, p=0.0000. Unstandardized coefficient $B_1$ shows that on average female have the higher life expectancy of 5.0617 units compared with their male counterpart. Similarly, model 2 reports that the magnitude of natural disaster significantly predicts the life expectancy, b=-1.2859, t(292)=-2.1073, p=0.0359. Unstandardized coefficient $B_2$ shows that for every 1 unit of increase in the magnitude of natural disaster, there is a 1.2859 unit of decrease in the life expectancy.

The same model also shows that women's socio-economic and political rights significantly predicts the life expectancy, b=2.5721, t(292)=2.0919, p=0.0373. Unstandardized coefficient $B_3$ shows that for every 1 unit of increase in the woman socio-economic and political rights, there are 2.5721 units of increase in the life expectancy.

In summary, findings show that the interaction effect between (1) the magnitude of natural disaster and gender; (2) magnitude of natural disaster and women's socio-economic and political rights; (3) gender and women's socio-economic and political rights; (4) the magnitude of natural disaster and gender and women's socio-economic and political rights are not significant, p>0.05. Thus, it can be concluded that gender and women's socio-economic and political rights do not individually affect the strength of the relationship between the magnitude of natural disaster and life expectancy.

To examine the effect of the magnitude of natural disaster on the life expectancy at each level of the gender and women's socio-economic and political rights, conditional effects are calculated. Table 7 reports the conditional effects of the magnitude of natural disaster on the life expectancy at the values of the first and second moderators. Table 7 indicates that magnitude of natural disaster does not significantly reduce the life expectancy of women at each level of their socio-economic and political rights: low (p=0.8964, p>0,05), average (p=0.2082, p>0,05), high (p=0.3419, p>0,05). Likewise, the magnitude of natural disaster does not significantly reduce the life expectancy of men at each level of the women's socio-economic and political rights: low (p=0.9573, p>0,05), average (p=0.0833, p>0,05), high (p=0.1539, p>0,05). Therefore, the third hypothesis is rejected.

**Hypothesis 3.** Magnitude of natural disasters reduce the life expectancy of women relatively more than that of men and this is more likely to increase in countries that are highly vulnerable to the impacts of climate- and nature-induced stressors

Three-way moderation analysis is used to test the fourth hypothesis. This examines whether the magnitude of natural disaster reduces the life expectancy of women relatively more than that of men and if this hypothesized phenomenon leads to an increase in a country that is highly vulnerable to the natural catastrophic shocks. In this analysis, the magnitude of natural

disaster is the independent variable, life expectancy is the dependent variable, gender is the moderator and the country's level of vulnerability to climate- and nature-induced stressors is the second moderator (see figure 4).

In table 6, model 3 shows that gender significantly predicts the life expectancy, b=5.0556, t(292)=12.8827, p=0.0000. Unstandardized coefficient $B_1$ shows that on average, the female has a higher life expectancy for 5.0556 units compared with their male counterpart. In addition, model 3 reports that the magnitude of natural disaster significantly does not predict the life expectancy, b=-.2599, t(292)=-.8359, p=0.4039. While the country's level of vulnerability to natural catastrophic shocks significantly predicts the life expectancy, b=-93.9752, t(292)=-24.2306, p=0.0000. Unstandardized coefficient $B_3$ indicates that for every 1 unit of increase in the country's level of vulnerability to climate- and nature-induced stressors, there are 93.9752 units of a decrease in the life expectancy.

Further, results of model 3 indicate that the interaction effect between (1) the magnitude of natural disaster and gender; (2) the magnitude of natural disaster and country's level of vulnerability to climate- and nature-induced stressors; (3) gender and country's level of vulnerability to climate- and nature-induced stressors; (4) the magnitude of natural disaster and gender and country's level of vulnerability to climate- and nature-induced stressors are not significant, p>0.05. Likewise, findings show that gender and country's level of vulnerability to climate- and nature-induced stressors do not affect the strength of the relationship between the magnitude of natural disaster and life expectancy.

To examine the effect of the magnitude of natural disaster on the life expectancy at each level of the gender and women's socio-economic and political rights, conditional effects are calculated. Table 7 reports the conditional effects of the magnitude of natural disaster on the life expectancy at the values of the first and second moderators. Table 7 indicates that magnitude of natural disaster does not significantly reduce the life expectancy of woman at each level of the women's socio-economic and political rights

There is no statistically significant effect of the magnitude of natural disaster on life expectancy for men (p=0.3174, p>0.05) and women (p=0.8737, p>0.05). Likewise, results in table 7 indicate that the magnitude of natural disaster has no statistically significant effect on the life expectancy for both women (p=0.5948, p>0.05) and men (p=0.2113, p>0.05) in a country that has a low level of vulnerability to climate- and nature-induced stressors. In fact, even in a country with high (men, p=0.8164, p>0.05; p=0.6082, p>0.05) and average level (men, p=0.3174, p>0.05; women, p=0.8737, p>0.05) of vulnerability to climate- and nature-induced stressors, the effects of the magnitude of natural disaster on men and women's life expectancy remain statistically insignificant. Based on this analysis, it can be concluded that the magnitude of natural disaster does not significantly reduce the life expectancy of both women and men at each level of country's vulnerability to the climate- and nature-induced stressors. Therefore, the fourth hypothesis is rejected.

**6. Conclusion**

This study examines the impact of natural disasters on the gender gap in life expectancy in Southeast Asia. It also sets out to test whether women's socio-economic and political rights and the country's vulnerability to climate- and nature-induced

stressors contribute to the hypothesized gendered impacts of natural disasters. The overall findings of the study are both in support and in opposition to some previous gender and natural disaster-related literature. I laid out statistical evidence to prove that the magnitude of natural disaster influences the gender gap in life expectancy in the region. This phenomenon provides support to the widespread perception that women are disproportionately affected by the impact of natural disaster compared to their male counterpart. However, other essential determinants – women's socio-economic and political rights and the vulnerability of the country to climate- and nature-induced stressors, do not provide strong support to the gendered differences and impacts of natural disasters on women population in Southeast Asia. These particular findings challenge those of other ecofeminist analyses and perceptions in which they articulate that gendered dimensions and disproportional features of natural disaster impacts are commonly associated with those mentioned socio-economic and political disadvantages present in most countries.

Using the set of data that measures women's socio-economic and political rights, we have quantitatively presented that there is no substantial evidence to support that such socio-economic and political developments provide significant influence to the gendered impact of natural disasters on women's life expectancy. Likewise, the measure of country's level of vulnerability to climate- and nature-induced stressors does not significantly influence the life expectancy of both women and men in the presence of the natural disaster. This particular finding is opposite to that of Neumayer and Plümper's (2007) popular study in which they concluded that women's mortality is more likely to be negatively affected by the strength of a natural disaster and a low socio-economic and political rights exacerbate the phenomenon.

Moreover, reports and studies argue that these disproportionate impacts of natural disasters extend to post-natural catastrophic events where women are more affected and targeted than men. In fact, several forms of sex and gender-based violence against women and girls in the aftermath of natural disaster occur in all countries and all phases of developments (Amnesty International, 2011; Le Masson, Lim, Budimir, & Podboj, 2016; IFRC, 2016). Types of violence include physical assault, rape, and sexual abuse of children. Therefore, there are other potential socioeconomic and political factors beyond the data we utilized in this study that could significantly link to the overall gendered impact of a natural disaster on women.

Although the study presented quantitative evidence to support that the impact of natural disaster is not gender neutral in Southeast Asia, it is worth noting that this study used the data sets of natural disaster-related deaths as an indicator of the strength of natural disaster. This measure is not reflective of the overall impact of natural disaster on the population at risk, particularly on women. One can argue that violence and some other form of inequalities in the society are categorized as the adverse impact of a natural catastrophic event.

In addition, one of the limitations of the study is its dependence on the measures of the impact of natural disasters and women's socio-economic and political rights. The availability of gender-disaggregated data on disaster-related deaths could potentially provide a more comprehensive result and draw a more relevant analysis on the actual impact of natural disasters on the gender gap in life expectancy, as well as the actual influence of other socio-economic and political determinants. Currently, the lack of sex-disaggregated data on natural disaster-related deaths hamper the research efforts of most

researcher. Future scholars and researchers are encouraged to explore other data sets available that represent other socio-economic, political, and cultural dimensions of the negative consequences of natural disaster.

**Table 1.** Summary statistics of natural disaster types in Southeast Asia (1995-2011)

| Disaster types | No. of occurrences | No. of deaths | No. of affected population |
|---|---|---|---|
| Drought | 21 | 680 | 38847602 |
| Earthquake | 64 | 184404 | 6564619 |
| Epidemic | 69 | 5785 | 297029 |
| Flood | 324 | 12416 | 91817750 |
| Insect infestation | 1 | 0 | 200 |
| Landslide | 64 | 3459 | 685673 |
| Mass movement (dry) | 1 | 11 | 0 |
| Storm | 203 | 156735 | 83284538 |
| Volcanic activity | 26 | 329 | 585829 |
| Wildfire | 15 | 245 | 34300 |
| **Total** | **788** | **364,064** | **222,117,540** |

Source: International Disaster Database (EM-DAT), 2017

**Table 2.** Summary of statistics of natural disaster occurrences and deaths

| Country | No. of occurrences | No. of deaths |
|---|---|---|
| **Brunei Darussalam** | 1 | 0 |
| **Cambodia** | 26 | 1558 |
| **Indonesia** | 214 | 184779 |
| **Laos** | 17 | 233 |
| **Malaysia** | 53 | 841 |
| **Myanmar** | 22 | 139359 |
| **Philippines** | 248 | 15611 |
| **Singapore** | 3 | 36 |
| **Thailand** | 78 | 11265 |
| **Timor-Leste** | 8 | 27 |
| **Vietnam** | 118 | 10355 |
| **Total** | **788** | **364,054** |

Source: International Disaster Database (EM-DAT), 2017

**Table 3.** Model summary: Hypothesis 1

| Model | R | R-square | Adjusted R-square | Change statistics | | | |
|---|---|---|---|---|---|---|---|
| | | | | R-square change | F change | df1 | df2 |
| 1 | 0.310a | 0.096 | 0.090 (0.0105) | 0.096 | 15.737** | 1 | 148 |
| 2 | 0.415b | 0.173 | 0.161 (0.0101) | 0.076 | 13.591** | 1 | 147 |

a. Predictors: (Constant). Magnitude of natural disaster (Death over population x 100.000)
b. Predictors: (Constant). Magnitude of natural disaster (Death over population x 100.000)
Notes: Magnitude of natural disaster x magnitude of natural disaster. Standard errors are in parenthesis

*p<0.05
**p<0.01

**Table 4.** ANOVA: Hypothesis 1

| | Model | Sum of Squares | Df | Mean square | F |
|---|---|---|---|---|---|
| 1 | Regression | 0.002 | 1 | 0.002 | 15.737 ** a |
| | Residual | 0.016 | 148 | 0.000 | |
| | Total | 0.018 | 149 | | |
| 2 | Regression | 0.003 | 2 | 0.002 | 15.334** b |
| | Residual | 0.015 | 147 | 0.000 | |
| | Total | 0.018 | 149 | | |

a. Predictors: (Constant), Magnitude of natural disaster (Death over population x 100.000)
b. Predictors: (Constant), Magnitude of natural disaster (Death over population x 100.000)
Notes: Dependent variable: Gender gap in life expectancy (female life expectancy over male's life expectancy). Magnitude of natural disaster x Magnitude of natural disaster

**p<0.01

**Table 5.** Unstandardized and standardized coefficients: Hypothesis 1

| Model | | Unstandardized coefficients | Standardized coefficients | T |
|---|---|---|---|---|
| | | **B** | **Beta** | |
| **1** | **(Constant)** | 1.075 | | 1083.405 |
| | | (0.001) | | |
| | **Magnitude of natural disaster** | 0.004** | 0.310 | 3.967 |
| | | (0.001) | | |
| **2** | **(Constant)** | 1.073 | | 990.878 |
| | | (0.001) | | |
| | **Magnitude of natural disaster** | 0.013** | 0.941 | 5.035 |
| | | (0.003) | | |
| | **Magnitude of natural disaster x magnitude of natural disaster** | 0.003** | -0.689 | -3.687 |
| | | (0.001) | | |

Notes: Dependent Variable: Gender gap in life expectancy (female life expectancy over male's life expectancy).
Standard errors are in parenthesis
**p<0.01

**Table 6.** Model summary and results: Hypothesis 2 & 3

|  | Model 2 | Model 3 |
| --- | --- | --- |
| **Constant** | 67.7314 | 67.7605 |
|  | (0.3971) | (0.1962) |
| **Gender** | 5.0617** | 5.0556** |
|  | (0.7941) | (0.3924) |
| **Magnitude of natural disaster** | -1.2859* | -0.2599 |
|  | (0.6102) | (0.3109) |
| **Magnitude of natural disaster x Gender** | 0.3187 | 0.3835 |
|  | (1.2204) | (0.6219) |
| **Women's socio-economic and political rights** | 2.5721* |  |
|  | (1.2296) |  |
| **Women's socio-economic and political rights x Magnitude of natural disaster** | -3.1438 |  |
|  | (2.7187) |  |
| **Women's socio-economic and political rights x Gender** | 0.2000 |  |
|  | (2.4592) |  |
| **Women's socio-economic and political rights x Gender x Magnitude of natural disaster** | 1.1077 |  |
|  | (5.4375) |  |
| **Country's level of vulnerability to climate- and nature-induced stressors** |  | -93.9752** |
|  |  | (3.8784) |
| **Country's level of vulnerability to climate- and nature-induced stressors x Magnitude of natural disaster** |  | 7.8030 |
|  |  | (6.8378) |

| | | |
|---|---|---|
| **Country's level of vulnerability to climate- and nature-induced stressors x Gender** | | -9.0414 (7.7567) |
| **Country's level of vulnerability to climate- and nature-induced stressors x Gender x Magnitude of natural disaster** | | -3.3784 (13.6756) |
| **Mean squared error** | 47.977 | 12.267 |
| **Number of observations** | 300 | 300 |
| **R-squared** | 0.1597 | 0.7851 |

Notes: The dependent variable is life expectancy, gender as the primary moderator, and two separate secondary moderators (i.e. women's socio-economic and political rights, and country's level of vulnerability to climate change stressors). The table reports coefficient values (b), and robust standard errors are in parentheses.
**p<0.01
*p<0.05

**Table 7.** Conditional effects and results: Hypothesis 2 & 3

| | Gender (primary moderator) | |
| --- | --- | --- |
| | **Male** | **Female** |
| **Women's socio-economic and political rights (second moderator)** | | |
| **Low** | -0.0569 | -0.1541 |
| | (1.0620) | (1.1826) |
| **Average** | -1.4453 | -1.1265 |
| | (0.8317) | (0.8932) |
| **High** | -2.8337 | -2.0990 |
| | (1.9820) | (2.2048) |
| **Country's level of vulnerability to climate- and nature-induced stressors (second moderator)** | | |
| **Low** | -1.0591 | -0.4594 |
| | (0.8454) | (0.8627) |
| **Average** | -0.4517 | -0.0681 |
| | (0.4509) | (0.4282) |
| **High** | 0.1558 | 0.3231 |
| | (0.6709) | (0.6296) |

Notes: The dependent variable is life expectancy, gender as the primary moderator, and two separate secondary moderators (i.e. women's socio-economic and political rights, and country's level of vulnerability to climate- and nature-induced stressors). The table reports conditional effects (b), and robust standard errors are in parentheses.

[Figure]

**Figure 1.** Hypothesis 1

**Gender gap in life expectancy**

[Figure]

**Magnitude of natural disaster**

**Figure 2.** Linear and quadratic model

[Figure]

**Figure 3.** Hypothesis 2

Country's level of
vulnerability to
climate- and nature-
induced stressors

Gender

Magnitude of
natural disaster

Life expectancy

**Figure 4.** Hypothesis 4

---

## Referee Comment (RC2) · K. Sudmeier-Rieux (Referee) · 2 Oct 2018

Review of the manuscript: Discovering the differential and gendered consequences of natural disasters on the gender gap in life expectancy in Southeast Asia

The manuscript topic is certainly worthy of attention and merits a lot more research than what is currently to be found in the literature. However unfortunately the manuscript falls short on a number of points: 1. It is not clear what is new about the research, the hypothesis and findings that will add to our understanding of gender and natural haz-

ards. The fact that women are more at risk and die during a natural hazard event has already been clearly established in the literature. The paper does not quote any major gaps that need to be addressed... rather they quote existing literature (esp Neumayer and Plumper, 2007) for several of their hypothesis. Perhaps the gap that should be articulated is a lack of quantitative evidence to which the study can add more quantitative proof ? Even there, I am not sure whether this is a gap.. 2. Since Neumayer and Plumper 2007, the literature on gender and disasters has moved beyond establishing women as victims but rather promotes them as 'agents of change' as part of disaster risk reduction programmes. This is where there are clear research gaps and a lack of ideas on how to advance this agenda. 3. Page 4 line 3 - I would avoid quoting statistics printed in the grey literature (Oxfam 2013). Best to go to the source - most likely EM-DAT, or other more reliable sources. 4. The authors go to great lengths to describe disasters as being human-induced (the jump from Rousseau (1756) to Pande (2000) is a bit much... and the World Bank publication, Unnatural disasters - should be quoted), therefore it is very astonishing that we still find the term 'natural disasters' throughout the manuscript - the term is mostly shunned by academia nowadays. 5. The authors also correlate high socio-economic levels and democracy with better preparedness and response to disasters - and this seems credible, with a few exceptions that should be mentioned, such as Cuba.. 6. Page 5, line 11. female casualties are evidently higher than men (not explained - why?) 7. Page 5, line 21 despite being restricted by... should be: in addition/ or because they are restricted 8. Merit should always be given to non-native speakers who publish in English - however the manuscript needs a lot more editing by a native speaker and quite a lot more refinement /sophistication in order to meet the high standard of NHESS articles.

---

## Author Comment (AC2) · 1 Nov 2018

All notes and comments are very well appreciated. Rest assured that I will keep all your pieces of advice in mind the next time I take on my subsequent research work. With that said, I decided to withdraw my submission and give myself some time to work on my research improvements.